# Postoperative Inpatient Rehabilitation Does Not Increase Knee Function after Primary Total Knee Arthroplasty

**DOI:** 10.3390/jpm12111934

**Published:** 2022-11-21

**Authors:** Dominik Rak, Alexander J. Nedopil, Eric C. Sayre, Bassam A. Masri, Maximilian Rudert

**Affiliations:** 1Orthopädische Klinik König-Ludwig-Haus, Lehrstuhl für Orthopädie der Universität Würzburg, 97074 Würzburg, Germany; 2Arthritis Research Canada, Vancouver, BC V5Y 3P2, Canada; 3Department of Orthopedic Surgery, University of British Columbia, Vancouver, BC V5Z 1M9, Canada

**Keywords:** total knee arthroplasty, fast track rehabilitation, inpatient rehabilitation, postoperative rehabilitation, patient reported outcome measures

## Abstract

Inpatient rehabilitation (IR) is a common postoperative protocol after total knee replacement (TKA). Because IR is expensive and should therefore be justified, this study determined the difference in knee function one year after TKA in patients treated with IR or outpatient rehabilitation, fast-track rehabilitation (FTR) in particular, which also entails a reduced hospital length of stay. A total of 205 patients were included in this multi-center prospective cohort study. Of the patients, 104 had primary TKA at a German university hospital and received IR, while 101 had primary TKA at a Canadian university hospital and received FTR. Patients receiving IR or FTR were matched by pre-operative demographics and knee function. Oxford Knee Score (OKS), Western Ontario and McMaster Universities Arthritis Index (WOMAC), and EuroQol visual analogue scale (EQ-VAS) determined knee function one year after surgery. Patients receiving IR had a 2.8-point lower improvement in OKS (*p* = 0.001), a 6.7-point lower improvement in WOMAC (*p* = 0.063), and a 12.3-point higher improvement in EQ-VAS (*p* = 0.281) than patients receiving FTR. IR does not provide long-term benefits to patient recovery after primary uncomplicated TKA under the current rehabilitation regime.

## 1. Introduction

Over the last two decades, the number of total joint replacement surgeries has consistently increased due to ageing in Western societies and growing numbers of people with overweight or obesity [1]. Due to the expected increase in patients needing a total knee arthroplasty (TKA), alternative rehabilitation pathways besides inpatient rehabilitation (IR) have been explored with the aim to provide comparable functional outcomes while reducing the time a patient is required to stay away from home.

IR after TKA is common practice in Western European countries. IR usually includes 3 weeks of medical rehabilitation at a rehabilitation center to reintroduce patients to independent living and social integration. Patients receive daily physical therapy, occupational therapy, and functional training. In addition, patients are provided accommodation and daily meals. After discharge from IR, outpatient physical therapy is continued for four more weeks.

Post-TKA fast-track rehabilitation (FTR) was developed with the goal of reducing costs and providing patients with a faster return to home while maintaining comparable functional outcomes. FTR entails a reduced length of stay in the hospital with discharge to home and not to IR. Outpatient physical therapy or inhouse visits by a physical therapist help patients to regain their knee function and return to an independent lifestyle, while forgoing IR can save a patient $3450 [2].

Because of the potential cost reduction, multiple countries have steered away from utilizing IR after uncomplicated TKAs [3]. One of these countries is Canada. In Germany, IR is still the standard postoperative rehabilitation regime after uncomplicated TKA. As insured patients in Germany would be reluctant to be randomized to a therapy other than the one to which they feel entitled, it is difficult to conduct a randomized controlled trial that compares a resource-intensive pathway (=IR) with a less resource-intensive alternative (=FTR). Consequently, the similarities between the Canadian and German healthcare systems offer an opportunity to compare functional improvements after these two different rehabilitation regimes [4].

Multiple studies—mainly performed outside Germany—have demonstrated equal or even improved patient joint function when FTR after TKA or total hip arthroplasty was compared to IR [2,5,6,7]. While German patients remain unwilling to forfeit their right to IR, a plausible method to compare the influence of IR and FTR on the functional improvement after TKA is to analyze PROMs between German patients receiving IR and patients from another country receiving FTR. To minimize the inherent bias when comparing patients from two different countries, the patients’ general health and pre-operative knee function and the surgical technique and implant design should be comparable. 

Accordingly, the purpose of the study was to compare the functional improvement after TKA among patients receiving IR in Germany with patients receiving FTR in Canada. The hypothesis was that FTR results in comparable functional improvement one year after TKA. A follow-up period of one year seemed reasonable, since knee function reaches a plateau within the first postoperative year, which remains stable in the following years [8,9,10,11]. A cost analysis, however, was not performed because of the countries’ different reimbursement policies. 

## 2. Materials and Methods

The study was planned and conducted as a transnational evaluation of prospectively collected data, in which the one-year clinical improvement after the Canadian FTR and the German IR following primary TKA was compared. The study was a mutual project of the orthopedic departments of University of British Columbia, Vancouver, Canada, and Julius Maximilian Universität, Würzburg, Germany. IR for the German patients took place at a preselected rehabilitation center. Canadian patients instead conducted outpatient physical therapy with corresponding home exercises or through home visits by a physical therapist.

Upon receipt of approval from both the Canadian (The University of British Columbia, Office of Research Ethics, Study ID: H18-02307, 05.09.2018) and German (Ethik-Kommission der Universität Würzburg, Number: 20210925 01, 26.20.2021) institutional ethics boards, we identified all patients between December 2019 and February 2020 who fulfilled the criteria for medical need for TKA treatment. Included were all patients with primary osteoarthritis and radiographic evidence of Kellgren–Lawrence grades II-IV osteoarthritic changes who underwent an uncomplicated unilateral TKA using the Stryker Triathlon TKA System. Patients receiving simultaneous bilateral TKAs, patients with previous fracture of the affected lower extremity, patients with metabolic or inflammatory joint disease (e.g., rheumatoid arthritis or osteonecrosis), and patients not following the standardized postoperative rehabilitation regime were not eligible to participate in the study. Patients lost to follow-up were excluded from the final data analysis.

During the above-mentioned time-period, a total of 117 TKAs were performed at the German hospital, of which the Stryker Triathlon System was used in 113. Nine patients were lost to follow-up, leaving 104 patients for final analysis. In Canada, 115 TKAs were performed in the above-mentioned time-period, all utilizing the Stryker Triathlon System; 14 patients were lost to follow-up, leaving 101 patients for final analysis (Table 1).

All TKAs were performed using a standard medial parapatellar approach, mechanical alignment principles, and a cruciate retaining femoral component with a cruciate retaining (CR) or cruciate-substituting polyethylene (CR-CS) Triathlon TKA system (Stryker, Kalamazoo, MI, USA).

The target hospital length-of-stay for the IR patients was 4–5 days before they were transferred to IR. IR was scheduled for three weeks and included daily physical therapy, occupational therapy, and functional training. In the FTR group, the target hospital length-of-stay was 1–2 days before home discharge and onset of outpatient rehabilitation. Outpatient rehabilitation included physical therapy twice per week for 6 weeks with instruction to perform independent exercises at home.

To compare the baseline health status between the two groups the pre-operative age, sex, body mass index (BMI), and American Society of Anesthesiologists (ASA) Physical Status score were obtained. 

To evaluate and assess knee function and the functional improvement after TKA the following patient-reported outcome measures (PROMs) were conducted. The Oxford Knee Score (OKS) is scored from 0 (worst) to 48 (best) and assesses knee function before and after TKA. It is self-conducted by the patient and comprises 12 questions that are divided into two subscales: pain and physical function. The Western Ontario and McMaster Universities Arthritis Index (WOMAC) is a self-administered questionnaire to assess pain, function, and stiffness of the knee. The descriptors range from no difficulty (0 point) to extreme difficulties (4 points). The WOMAC Score is a commonly used standardized questionnaire to evaluate the condition of patients with osteoarthritis of the knee and hip. The WOMAC is scored from 0 (best) to 96 (worst). Furthermore, the EuroQol visual analogue scale (EQ-VAS), which measures health-related quality of life, was completed by patients. It is scored from 0 (worst to 100 (best) and quantifies the patient’s perception of their health state. It is self-conducted by the patient and has 5 different dimensions: mobility, self-care, usual activities, pain/discomfort, and anxiety/depression.

The difference between the OKS, WOMAC, and EQ-VAS obtained pre-operatively and at the one-year follow-up visit defined clinical improvement. The minimum clinically important difference (MCID) for OKS was 5.0, for WOMAC, 10, and for EQ-VAS, 8.0 [12,13,14].

### Statistical Analysis

In unadjusted analyses, IR and FTR were compared with pre-operative age, sex, BMI, ASA, OKS, WOMAC, and EQ-5D VAS. Binary or categorical comparisons were made via exact chi-squared tests, while continuous comparisons were made via the Wilcoxon rank-sum test. Adjusted multivariable linear regression models were fitted to predict improvements from pre-operative to follow-up in OKS, WOMAC, and EQ-5D VAS vs. rehabilitation protocol, adjusting for age, sex, BMI, ASA, and the pre-operative value of the PROM being analyzed. Fit of linear regression models were assessed via normal quantile–quantile plots of the standardized residuals. Fit of logistic regression models were assessed with the Hosmer and Lemeshow goodness-of-fit test at alpha = 0.05 [15]. SAS software (version 9.4, SAS Institute Inc., Cary, NC, USA) was used for statistical analyses.

## 3. Results

A total of 205 patients participated in the study with available PROMs at 1-year follow-up. Patient pre-operative demographics and knee function were comparable between the FTR and IR groups (Table 1 and Table 2).

Patients receiving IR had a significantly lower mean improvement in OKS (14.0 ± 10.7) and a trend of lower improvement in WOMAC (−26.5 ± 25.5) than patients receiving FTR (OKS 16.8 ± 10.6; WOMAC −33.2 ± 20.6), resulting in adjusted models for *p*-values of 0.001 and 0.063, respectively. The EQ-VAS tended to improve more after IR (24.2 ± 26.9) than after FTR (11.9 ± 24.2) (*p* = 0.281) (Table 2 and Figure 1). All differences in follow-up OKS, WOMAC, and EQ-VAS between the FTR and IR groups were below the MCID. 

## 4. Discussion

The most important finding from the present study was that FTR after TKA did not result in inferior functional improvement when compared to IR. Indeed, patients receiving FTR showed higher improvement in PROMs than patients receiving IR. While outcomes scores at one year were higher in the FTR group regardless of the degree of improvement, this difference was not clinically significant. These results may encourage insurance companies to recommend FTR to their patients, as FTR does not compromise patient outcome.

These findings agree with multiple international studies comparing outpatient and inpatient rehabilitation [4,7]. Prospective and retrospective studies, mainly from outside Europe, demonstrated comparable or even improved function upon utilizing outpatient rehabilitation after TKA, with reduced treatment costs [2,4,16,17,18,19]. As the risk of complications seemed not to increase, the conclusion of these studies was predominantly to not recommend IR to patients undergoing uncomplicated TKA. A dissertation conducted at the University of Rostock/Germany comparing outpatient and inpatient rehabilitation was in agreement with these findings [5]. Unfortunately, the dissertation has not yet been published in the medical literature. Because the improvements in PROMs in the present study were adjusted for age, sex, ASA, and the pre-operative value of the corresponding PROM, the observed differences in improvements being influenced by these variables can be ruled out. I.e., irrespective of the patients’ age, sex, ASA, and pre-operatively assessed PROM, the improvement in PROMs is comparable or slightly higher after FTR than IR. While comparison of clinical improvement after FTR or IR can be confounded by a selection bias, i.e., patients receiving FTR could be fitter, healthier, and more motivated to improve than patients receiving IR, the results from this study are not biased by a selection process because both patient groups followed the standard postoperative protocol of the treating hospital [17]. 

In addition to providing comparable functional recovery, FTR has also been shown to reduce costs [2]. These cost reductions include costs associated with the necessary clinical wound check and staple removal follow-up of patients not receiving IR. Obviously, not providing patients with accommodation, food, and sanitary needs reduces costs. Outpatient rehabilitation also reduces readmission rate after TKA and the risk of peri-prosthetic complications [7,18]. The number of diagnostic tests can be reduced with no negative effect on patient outcome [19]. Because FTR includes a reduced length of hospital stay, it can be expected that costs can be further reduced using this rehabilitation protocol [20].

Because this study favors FTR due to comparable or slightly higher functional improvement one year after TKA, it could motivate carriers of IR facilities to reorient the focus of their rehabilitation regime. One potential reason that patients after FTR achieve higher improvement than after IR is the self-guidance and patient responsibility to regain their independence and function. The self-guidance potentially improves function in two ways. (1) It fosters salutogenesis by focusing the patient towards solving problems after TKA, which include regaining range of motion, mobility, and strength [21]. (2) It prevents externalization of the rehabilitation process, i.e., success or failure to improve is not the physical therapist’s responsibility, it is the patient’s responsibility.

There are some limitations to this study. First, the difference in clinical improvement could be influenced by country-specific pain awareness of patients and their interpretation of functional limitations. Because the pre-operative OKS and WOMAC were not different between the two countries and because the comparison of clinical improvement was normalized to the pre-operative PROM, the country-specific influence on clinical improvement should be minimal. Second, because patients were treated by two different orthopedic surgeon teams in different countries, multiple variables can influence patient clinical improvement besides the rehabilitation protocol. To minimize the influence of other variables, we ensured that surgical technique, component alignment target, and component design were identical between patient groups. Consequently, only a single hospital participated in Germany and in Canada, which could limit the generalizability of the results. Still, these were large teaching hospitals with high volumes, and as such were representative for the comparison of the two different rehabilitation protocols. Third, a cost analysis and comparison between both rehabilitation protocols could not be made, since profoundly different reimbursement policies were used. However, the gained knowledge from this study that functional gains after TKA are not compromised by omitting IR could motivate German insurance carriers, hospital systems, and healthcare authorities to conduct prospective studies analyzing the financial impact of omitting IR after uncomplicated TKA. Fourth, shorter follow-up intervals within the first year after TKA could have provided a more detailed comparison between FTR and IR [10]. Ideally, this comparison can be performed in a prospective study within Germany, which will also allow an accurate cost analysis that includes the financial impact of patient sick leave. Finally, the results from this study only apply to patients undergoing primary TKA with an ASA of I to III. 

## 5. Conclusions

Among adults undergoing primary TKA, the use of IR compared with FTR does not yield a higher improvement in knee function. These findings do not support IR for this group of patients with the current rehabilitation regime. This study can provide a thought-provoking impulse to IR carriers to reorient their focus on rehabilitation after TKA by fostering patient salutogenesis.

## Figures and Tables

**Figure 1 jpm-12-01934-f001:**
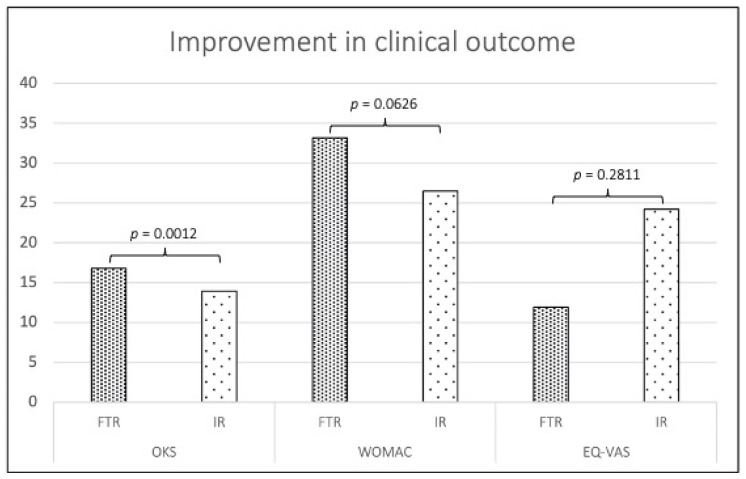
Improvement in clinical outcome. OKS, Oxford Knee Score; WOMAC, Western Ontario and McMaster Universities Arthritis Index; EQ-VAS, EuroQol visual analogue scale; FTR, fast-track rehabilitation; IR, inhouse rehabiliation; *p*-values for clinical improvements are from multivariable models; improvements in WOMAC are shown in the positive direction for comparison.

**Table 1 jpm-12-01934-t001:** Patient pre-operative demographics.

Rehabilitation Protocol		FTR-Canada (n = 101)	IR-Germany (n = 104)	*p*-Value
**Age (years)** mean ± SD		66 ± 8	67 ± 10	0.703
**Sex**	**Female (n) (%)**	65 (64%)	57 (55%)	0.205
	**Male (n) (%)**	36 (36%)	47 (45%)	
**BMI** mean ± SD		30.1 ± 6.4	31.6 ± 6.0	0.029
**ASA**				0.189
	**I (n) (%)**	9 (9%)	4 (4%)	
	**II (n) (%)**	71 (70%)	70 (67%)	
	**III (n) (%)**	21 (21%)	30 (29%)	

BMI, body mass index; ASA, American Society of Anesthesiologists Physical Status; SD, standard deviation; FTR, fast-track rehabilitation; IR, inhouse rehabilitation.

**Table 2 jpm-12-01934-t002:** Clinical scores in both groups.

	Rehabilitation Protocol	Pre-Operative Mean ± SD	*p*-Value	1-Year Follow-Up Mean ± SD	*p*-Value	Clinical Improvement ∆ Mean ± SD	*p*-Value
OKS	FTR	21.9 ± 8.4	0.111	38.7 ± 8.8	<0.001	16.8 ± 10.6	0.001
	IR	19.9 ± 6.6		33.9 ± 9.9		14.0 ± 10.7	
WOMAC	FTR	47.7 ± 20.1	0.177	14.7 ± 15.2	0.571	33.2 ± 20.6	0.063
	IR	44.8 ± 19.8		18.3 ± 18.7		26.5 ± 25.5	
EQ-VAS	FTR	66.4 ± 19.6	<0.001	78.6 ± 17.6	0.103	11.9 ± 24.2	0.281
	IR	48.4 ± 17.3		72.6 ± 22.1		24.2 ±26.9	

OKS, Oxford Knee Score; WOMAC, Western Ontario and McMaster Universities Arthritis Index; EQ-VAS, EuroQol visual analogue scale; FTR, fast-track rehabilitation; IR, inhouse rehabiliation; *p*-values for clinical improvements are from multivariable models; improvements in WOMAC are shown in the positive direction for comparison.

## Data Availability

The data sets to support the findings of this study are included within the article, including figures and tables. Any other data used to support the findings of this study are available from the corresponding authors upon request.

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
