# Peer review of "Postoperative Inpatient Rehabilitation Does Not Increase Knee Function after Primary Total Knee Arthroplasty"

_jpm, 2022, doi:10.3390/jpm12111934_

Round 1
Reviewer 1 Report
It is an interesting study. The authors compared the effect of functional recovery in patients after TKA who underwent rehabilitation with IR or FTR for one year. The results showed no significant difference in knee function between patients who received IR or FTR after one year. Furthermore, patients who received FTR had lower medical costs and reduced length of hospital stay compared to IR. The results of this study could inform government health policies and insurance company services. These are all very meaningful findings. But this study still has some problems and confusions that need to be revised.
1. The authors did not explain the specifics of IR and FTR. The IR and FTR mentioned in the article did not specify the length of rehabilitation for IR and the frequency of rehabilitation for FTR. The length and frequency of rehabilitation could affect the outcome of postoperative recovery and the cost of treatment for rehabilitation.
2. The data in the article only collected and compared after one year of rehabilitation treatment, which could not give a more detailed information about the functional recovery. For both IR and TFR rehabilitation modalities, the time required for patients to regain function may be different, which may also affect the cost of rehabilitation.
3. The study lacked data support for the comparison of rehabilitation costs and may require the collection of relevant data for analysis. Moreover, this study was designed for two countries, and there might be differences in health systems and insurance systems between countries, so objective data were needed to support the argument.
4. The inclusion criteria for patients were not clearly explained in the study.
Author Response
Review 1:
Dear reviewer,
We have read your comments carefully and appreciate your suggestions. We have tried to thoroughly edit and update our article as shown below:
Comment 1:
Introduction:
The authors did not explain the specifics of IR and FTR. The IR and FTR mentioned in the article did not specify the length of rehabilitation for IR and the frequency of rehabilitation for FTR. The length and frequency of rehabilitation could affect the outcome of postoperative recovery and the cost of treatment for rehabilitation.
Authors’ response:
We updated the specifics of the two different rehabilitation protocols in the manuscript as follows:
INTRODUCTION:
Line 29-34: IR after TKA is common practice in Western European countries. IR usually includes 3 weeks of medical rehabilitation at a rehabilitation center to reintroduce patients to independent living and social integration. Patients receive daily physical therapy, occupational therapy, and functional training. In addition, patients are provided accommodation and daily meals. After discharge from IR, outpatient physical therapy is continued for four more weeks.
METHODS:
Line 96-101: The target hospital length-of-stay for the IR patients was 4-5 days before they were transferred to IR. IR was scheduled for three weeks and included daily physical therapy, occupational therapy, and functional training. In the FTR group the target hospital length-of-stay was 1-2 days before home discharge and onset of outpatient rehabilitation. Outpatient rehabilitation included physical therapy twice per week for 6 weeks with instruction to perform independent exercises at home.
Comment 2:
The data in the article only collected and compared after one year of rehabilitation treatment, which could not give a more detailed information about the functional recovery. For both IR and TFR rehabilitation modalities, the time required for patients to regain function may be different, which may also affect the cost of rehabilitation.
Authors’ response:
We agree with the reviewer’s concern that there is a multitude of factors influencing functional outcome after TKA including time. Consequently, we added references and the following paragraph to the Introduction and Discussion to explain the rational for a one-year follow up.
INTRODUCTION:
Line 61-62: A follow-up period of one year seemed reasonable, since knee function reaches a plateau within the first postoperative year, which remains stable in the following years [1-4].
DISCUSSION:
Line 219-222: Fourth, shorter follow-up intervals within the first year after TKA could have provided a more detailed comparison between FTR and IR [3]. Ideally, this comparison can be performed in a prospective study within Germany, which will also allow an accurate cost analysis including the financial impact of patients’ sick leave.
Comment 3:
The study lacked data support for the comparison of rehabilitation costs and may require the collection of relevant data for analysis. Moreover, this study was designed for two countries, and there might be differences in health systems and insurance systems between countries, so objective data were needed to support the argument.
Authors’ response:
The reviewer correctly pointed out that a comparison of costs between the two countries might not do justice to the true difference in treatment costs. While FTR is motivated by cost-reductions this manuscript aims to answer the question whether patient’s function suffers from withholding IR. We clarified the aim in the manuscript as follows:
INTRODUCTION:
Line 63-64: A cost analysis, however, was not performed because of the countries’ different reimbursement policies.
DISCUSSION:
Line 213-229: Third, a cost analysis and comparison between both rehabilitation protocols could not be made since profoundly different reimbursement policies were used. However, the gained knowledge from this study that functional gains after TKA are not compromised by omitting IR could motivate German insurance carriers, hospital systems, and health care authorities to conduct prospective studies analyzing the financial impact of omitting IR after uncomplicated TKA.
Comment 4:
The inclusion criteria for patients were not clearly explained in the study.
Authors’ response:
Regarding the inclusion criteria we updated the materials and methods section as follows:
Line 78-91: Included were all patients with primary osteoarthritis and radiographic evidence of Kellgren-Lawrence grades II-IV osteoarthritic changes who underwent an uncomplicated unilateral TKA using the Stryker Triathlon TKA System. Patients receiving simultaneous bilateral TKAs, patients with previous fracture of the affected lower extremity, patients with metabolic or inflammatory joint disease (e.g., rheumatoid arthritis or osteonecrosis), and patients not following the standardized postoperative rehabilitation regime were not eligible to participate in the study. Patients lost to follow-up were excluded from the final data analysis.
During the above-mentioned time period a total of 117 TKAs were performed at the German hospital, of which the Stryker Triathlon System was used in 113 TKAs. Nine patients were lost to follow-up leaving 104 patients for final analysis. In Canada, 115 TKAs were performed in the above-mentioned time period, all utilizing the Stryker Triathlon System. 14 patients were lost to follow-up leaving 101 patients for final analysis (Table 1).
- Scott CEH, Bell KR, Ng RT, MacDonald DJ, Patton JT, Burnett R. Excellent 10-year patient-reported outcomes and survival in a single-radius, cruciate-retaining total knee arthroplasty. Knee Surg Sports Traumatol Arthrosc 27(4): 1106, 2019
- Cosendey K, Eudier A, Fleury N, Pereira LC, Favre J, Jolles BM. Ten-year follow-up of a total knee prosthesis combining multi-radius, ultra-congruency, posterior-stabilization and mobile-bearing insert shows long-lasting clinically relevant improvements in pain, stiffness, function and stability. Knee Surgery, Sports Traumatology, Arthroscopy, 2022
- Fransen BL, Hoozemans MJ, Argelo KD, Keijser L, Burger BJ. Fast-track total knee arthroplasty improved clinical and functional outcome in the first 7 days after surgery: a randomized controlled pilot study with 5-year follow-up. Archives of orthopaedic and trauma surgery 138(9): 1305, 2018
- Seetharam A, Deckard ER, Ziemba-Davis M, Meneghini RM. The AAHKS clinical research award: are minimum two-year patient-reported outcome measures necessary for accurate assessment of patient outcomes after primary total knee arthroplasty? The Journal of Arthroplasty, 2022
Reviewer 2 Report
Dear Authors, I have read with interest your article comparing functional improvement after total knee replacement between patients receiving hospital rehabilitation and patients receiving accelerated rehabilitation. I think some changes are needed which I list below: INTRODUCTION - Outpatient physical therapy or inhouse visits by a physical therapist replace the tasks of IR… .. clarify this concept better - The Oxford-Knee-Score (OKS) is scored from 0 (worst) to 48 (best) and assesses knee function before and after TKA. The Western Ontario and McMaster Univer sities Arthritis Index (WOMAC) is scored from 0 (best) to 96 (worst). The WOMAC is a self-administered questionnaire to assess pain, function, and stiffness of the knee. The de scriptors range from no difficulty (0 point) to extreme difficulties (4 points). The EuroQol visual analogue scales (EQ-VAS) (0 worst, 100 best) quantifies the patient's perception of health state ... ..I would post this whole paragraph in the materials and methods, dedicating a few lines to each evaluation scale used. - Expand the introduction by focusing on the problems that led to your research METHODS - Add additional inclusion criteria - A paragraph should be devoted to the type of rehabilitation the patients underwent in this study DISCUSSION - Expand, comparing with other data in the literature
Author Response
Review 2
Dear reviewer,
We have read your comments carefully and appreciate your suggestions. We have tried to thoroughly edit and update our article as shown below:
Comment 1:
INTRODUCTION - Outpatient physical therapy or inhouse visits by a physical therapist replace the tasks of IR… .. clarify this concept better
Authors’ response:
The sentence was edited as follows:
INTRODUCTION:
Line 38-40: Outpatient physical therapy or inhouse visits by a physical therapist help patients to regain their knee function and return to an independent lifestyle.
Comment 2:
The Oxford-Knee-Score (OKS) is scored from 0 (worst) to 48 (best) and assesses knee function before and after TKA. The Western Ontario and McMaster Univer sities Arthritis Index (WOMAC) is scored from 0 (best) to 96 (worst). The WOMAC is a self-administered questionnaire to assess pain, function, and stiffness of the knee. The de scriptors range from no difficulty (0 point) to extreme difficulties (4 points). The EuroQol visual analogue scales (EQ-VAS) (0 worst, 100 best) quantifies the patient's perception of health state ... ..I would post this whole paragraph in the materials and methods, dedicating a few lines to each evaluation scale used.
Authors’ response:
We appreciate the reviewer’s suggestion and updated the section and added the description of the PROMS to „Materials and methods“ section as follows:
Line 105-118: To evaluate and assess knee function and the functional improvement after TKA the following patient-reported outcome measures (PROMs) were conducted. The Oxford-Knee-Score (OKS) is scored from 0 (worst) to 48 (best) and assesses knee function before and after TKA. It is self-conducted by the patient and comprises 12 questions which are divided in two subscales pain and physical function. The Western Ontario and McMaster Universities Arthritis Index (WOMAC) is a self-administered questionnaire to assess pain, function, and stiffness of the knee. The descriptors range from no difficulty (0 point) to extreme difficulties (4 points). The WOMAC Score is a commonly used standardized questionnaire to evaluate the condition of patients with osteoarthritis of the knee and hip. The WOMAC is scored from 0 (best) to 96 (worst). Furthermore the EuroQol-visual analogue scales (EQ-VAS) was conducted by the patients which measures the health-related quality of life. It is scored from 0 (worst to 100 (best) and quantifies the patient’s perception of health state. It is self-conducted by the patient and has 5 different dimensions: mobility, self-care, usual activities, pain/discomfort and anxiety/depression.
Comment 3:
Expand the introduction by focusing on the problems that led to your research METHODS
Authors’ response:
The reviewer’s request has helped us focus the introduction and funnel the reader’s attention to the problem.
Line 23-49: Over the last two decades the number of total joint replacement surgeries increased consistently due to ageing Western societies as well as growing numbers of people with overweight or obesity [1]. Due to the expected increase in patients needing a total knee arthroplasty (TKA), alternative rehabilitation pathways besides inpatient rehabilitation (IR) have been explored with the aim to provide comparable functional outcomes while reducing the time a patient is required to stay away from home.
IR after TKA is common practice in Western European countries. IR usually includes 3 weeks of medical rehabilitation at a rehabilitation center to reintroduce patients to independent living and social integration. Patients receive daily physical therapy, occupational therapy, and functional training. In addition, patients are provided accommodation and daily meals. After discharge from IR, outpatient physical therapy is continued for four more weeks.
With the goal of reducing costs, providing patients with a faster return to home, while maintaining comparable functional outcomes, the concept of fast-track rehabilitation (FTR) after TKA has been developed. FTR entails a reduced length of stay in the hospital with discharge to home and not to IR. Outpatient physical therapy or inhouse visits by a physical therapist help patients to regain their knee function and return to an independent lifestyle. To forgo IR can save $3,450 [2].
Because of the potential cost reduction, multiple countries have steered away from utilizing IR after uncomplicated TKAs [3]. One of these countries is Canada. In Germany IR is still the standard postoperative rehabilitation regime after uncomplicated TKA. As insured patients in Germany would be reluctant to be randomized to a therapy other than the one, they feel entitled to, it is difficult to conduct a randomized controlled trial that compares a resource-intensive pathway (=IR) with a less resource-intensive alternative (=FTR). Consequently, the similar Canadian and German health care systems offers an opportunity to compare functional improvements after these two different rehabilitation regimes [4].
Comment 4:
Add additional inclusion criteria
Authors’ response:
Regarding the inclusion criteria we updated the materials and methods section as follows:
Line 78-91: Included were all patients with primary osteoarthritis and radiographic evidence of Kellgren-Lawrence grades II-IV osteoarthritic changes who underwent an uncomplicated unilateral TKA using the Stryker Triathlon TKA system. Patients receiving simultaneous bilateral TKAs, patients with previous fracture of the affected lower extremity, patients with metabolic or inflammatory joint disease (e.g., rheumatoid arthritis or osteonecrosis), and patients not following the standardized postoperative rehabilitation regime were not eligible to participate in the study. Patients lost to follow-up were excluded from the final data analysis.
During the above-mentioned time period a total of 117 TKAs were performed at the German hospital, of which the Stryker Triathlon System was used in 113 TKAs. Nine patients were lost to follow-up leaving 104 patients for final analysis. In Canada, 115 TKAs were performed in the above-mentioned time period, all utilizing the Stryker Triathlon System. 14 patients were lost to follow-up leaving 101 patients for final analysis (Table 1).
Comment 5:
A paragraph should be devoted to the type of rehabilitation the patients underwent in this study
Authors’ response:
We updated the specifics of the two different rehabilitation protocols in the manuscript as follows:
INTRODUCTION:
Line 29-34: IR after TKA is common practice in Western European countries. IR usually includes 3 weeks of medical rehabilitation at a rehabilitation center to reintroduce patients to independent living and social integration. Patients receive daily physical therapy, occupational therapy, and functional training. In addition, patients are provided accommodation and daily meals. After discharge from IR, outpatient physical therapy is continued for four more weeks.
METHODS:
Line 96-101: The target hospital length-of-stay for the IR patients was 4-5 days before they were transferred to IR. IR was scheduled for three weeks and included daily physical therapy, occupational therapy, and functional training. In the FTR group the target hospital length-of-stay was 1-2 days before home discharge and onset of outpatient rehabilitation. Outpatient rehabilitation included physical therapy twice per week for 6 weeks with instruction to perform independent exercises at home.
Comment 6:
DISCUSSION - Expand, comparing with other data in the literature
Authors’ response:
As suggested by the reviewer we expanded the discussion by a more detailed comparison to current literature:
Line 166-174: These findings agree with multiple international studies comparing outpatient and inpatient rehabilitation [4, 5]. Prospective and retrospective studies, mainly from outside Europe, demonstrated comparable or even improved function utilizing outpatient rehabilitation after TKA, with reduced treatment costs [2, 4, 6-9]. As the risk of complications seemed not to increase the conclusion of these studies was predominantly to not recommend IR to patients undergoing uncomplicated TKA. A dissertation conducted at the University of Rostock/Germany comparing outpatient and inpatient rehabilitation was in agreement with these findings [10]. Unfortunately, the dissertation has not been published in the medical literature, yet.
- Jiang L, Rong J, Wang Y, Hu F, Bao C, Li X, Zhao Y. The relationship between body mass index and hip osteoarthritis: a systematic review and meta-analysis. Joint, bone, spine : revue du rhumatisme 78(2): 150, 2011
- Mahomed NN, Davis AM, Hawker G, Badley E, Davey JR, Syed KA, Coyte PC, Gandhi R, Wright JG. Inpatient compared with home-based rehabilitation following primary unilateral total hip or knee replacement: a randomized controlled trial. J Bone Joint Surg Am 90(8): 1673, 2008
- Burnett III RA, Serino J, Yang J, Della Valle CJ, Courtney PM. National trends in post-acute care costs following total knee arthroplasty from 2007 to 2016. The Journal of Arthroplasty 36(7): 2268, 2021
- Naylor JM, Hart A, Mittal R, Harris I, Xuan W. The value of inpatient rehabilitation after uncomplicated knee arthroplasty: a propensity score analysis. Medical Journal of Australia 207(6): 250, 2017
- Onggo JR, Onggo JD, De Steiger R, Hau R. The Efficacy and Safety of Inpatient Rehabilitation Compared With Home Discharge After Hip or Knee Arthroplasty: A Meta-Analysis and Systematic Review. The Journal of Arthroplasty 34(8): 1823, 2019
- Buhagiar MA, Naylor JM, Harris IA, Xuan W, Kohler F, Wright R, Fortunato R. Effect of Inpatient Rehabilitation vs a Monitored Home-Based Program on Mobility in Patients With Total Knee Arthroplasty: The HIHO Randomized Clinical Trial. Jama 317(10): 1037, 2017
- Chan HY, Sultana R, Yeo SJ, Chia SL, Pang HN, Lo NN. Comparison of outcome measures from different pathways following total knee arthroplasty. Singapore Med J 59(9): 476, 2018
- Jorgenson ES, Richardson DM, Thomasson AM, Nelson CL, Ibrahim SA. Race, Rehabilitation, and 30-Day Readmission After Elective Total Knee Arthroplasty. Geriatr Orthop Surg Rehabil 6(4): 303, 2015
- White PB, Carli AV, Meftah M, Ghazi N, Alexiades MM, Windsor RE, Ranawat AS. Patients Discharged to Inpatient Rehabilitation Facilities Undergo More Diagnostic Interventions With No Improvement in Outcomes. Orthopedics 41(6): E841, 2018
- El-Aarid N. Effektivität von stationärer und ambulanter Rehabilitation bei Patienten nach Knieendoprothesen-Implantation. In.: Universität Rostock. 2019
Round 2
Reviewer 2 Report
I thank the authors for making changes to the manuscript following my suggestions. I consider the article to be acceptable in the following form.